# Machine Learning Model to Stratify the Risk of Lymph Node Metastasis for Early Gastric Cancer: A Single-Center Cohort Study

**DOI:** 10.3390/cancers14051121

**Published:** 2022-02-22

**Authors:** Ji-Eun Na, Yeong-Chan Lee, Tae-Jun Kim, Hyuk Lee, Hong-Hee Won, Yang-Won Min, Byung-Hoon Min, Jun-Haeng Lee, Poong-Lyul Rhee, Jae J. Kim

**Affiliations:** 1Department of Medicine, Samsung Medical Center, Sungkyunkwan University School of Medicine, Seoul 06351, Korea; jieun90.na@samsung.com (J.-E.N.); yangwon.min@samsung.com (Y.-W.M.); lamsu.min@samsung.com (B.-H.M.); jh2145.lee@samsung.com (J.-H.L.); pl.rhee@samsung.com (P.-L.R.); jaej.kim@samsung.com (J.J.K.); 2Department of Medicine, Inje University Haeundae Paik Hospital, Busan 48108, Korea; 3Department of Digital Health, Samsung Advanced Institute for Health Sciences & Technology (SAIHST), Sungkyunkwan University of Medicine, Seoul 06351, Korea; conan_8th@naver.com (Y.-C.L.); wonhh@skku.edu (H.-H.W.)

**Keywords:** early gastric cancer, machine learning model, risk stratification, lymph node metastasis

## Abstract

**Simple Summary:**

Endoscopic resection (ER) is a treatment option for clinically T1a early gastric cancer (EGC) without suspicion of lymph node metastasis (LNM). In patients with non-curative resection after ER, additional surgery is recommended owing to the LNM risk. However, of those patients treated with additional surgery after ER, the actual rate of LNM was about 5–10%; that is, the other patients underwent unnecessary surgeries. Therefore, it is crucial to estimate LNM risk in EGC patients to determine additional management after ER. We derived a machine learning (ML) model to stratify the LNM risk in EGC patients and validate its performance. The constructed ML model, which showed good performance with an area under the receiver operating characteristic of 0.85 or higher, could stratify LNM risk into very low (<1%), low (<3%), intermediate (<7%), and high (≥7%) risk categories. These findings suggest that the ML model can stratify the LNM risk in EGC patients.

**Abstract:**

Stratification of the risk of lymph node metastasis (LNM) in patients with non-curative resection after endoscopic resection (ER) for early gastric cancer (EGC) is crucial in determining additional treatment strategies and preventing unnecessary surgery. Hence, we developed a machine learning (ML) model and validated its performance for the stratification of LNM risk in patients with EGC. We enrolled patients who underwent primary surgery or additional surgery after ER for EGC between May 2005 and March 2021. Additionally, patients who underwent ER alone for EGC between May 2005 and March 2016 and were followed up for at least 5 years were included. The ML model was built based on a development set (70%) using logistic regression, random forest (RF), and support vector machine (SVM) analyses and assessed in a validation set (30%). In the validation set, LNM was found in 337 of 4428 patients (7.6%). Among the total patients, the area under the receiver operating characteristic (AUROC) for predicting LNM risk was 0.86 in the logistic regression, 0.85 in RF, and 0.86 in SVM analyses; in patients with initial ER, AUROC for predicting LNM risk was 0.90 in the logistic regression, 0.88 in RF, and 0.89 in SVM analyses. The ML model could stratify the LNM risk into very low (<1%), low (<3%), intermediate (<7%), and high (≥7%) risk categories, which was comparable with actual LNM rates. We demonstrate that the ML model can be used to identify LNM risk. However, this tool requires further validation in EGC patients with non-curative resection after ER for actual application.

## 1. Introduction

Early gastric cancer (EGC) describes a gastric tumor confined to the submucosa with or without lymph node metastasis (LNM). Endoscopic resection (ER) is recommended as a minimally invasive treatment for clinically mucosal EGC without suspicion of LNM [1,2,3,4]. In cases of non-curative resection after ER that do not satisfy the expanded criteria of curative resection, additional surgery is recommended, considering the risk of LNM [5,6]; however, LNM is found in only 5–10% of those patients after surgery [7,8,9,10]. Therefore, overtreatment is a concern. To address this, the recently revised guidelines excluded piecemeal resection and a positive lateral margin from the factors of non-curative resection after ER for which additional surgery is primarily recommended [1,4,11].

Furthermore, in Japan, patients who have non-curative resection after ER, excluding piecemeal resection and a positive lateral margin, are classified as “endoscopic curability (eCura) C-2”; patients in the eCura C-2 category are further stratified into low (2.5%), intermediate (6.7%), and high (22.7%) LNM risk categories based on the eCura scoring system [2,12,13]. In the low-risk category, there is no difference in cancer recurrence or cancer-specific mortality between patients who undergo no additional treatment and those who undergo additional surgery [14]. Hence, this LNM risk stratification system suggests that additional surgery after non-curative resection may be determined on an individual basis, considering the LNM risk, the patient’s condition, and the benefits and limitations of additional surgery [11,12,14].

Another area of concern is that some patients who were confirmed non-curative resection after ER without actual LNM may be unnecessarily exposed to surgery-related risks. The rates of postoperative complications and overall mortality after gastric cancer surgery are 10–26% and 0.3–2.3%, respectively, and comorbidities, body mass index, and lymph node dissection have been reported as risk factors [15,16,17,18,19,20,21]. In addition, the potential for long-term health problems after gastric cancer surgery, such as reflux, gastroparesis, gallstone, and osteoporosis, must be considered [22,23]. Therefore, it is clinically significant to predict the LNM risk among EGC patients who undergo non-curative resection after ER to prevent unnecessary surgery.

To stratify the LNM risk in EGC patients, we created a machine learning (ML) model for predicting LNM risk and validated its performance.

## 2. Materials and Methods

### 2.1. Patients

We included patients who underwent surgery for EGC between May 2005 and March 2021 at Samsung Medical Center. Additionally, patients who underwent additional surgery after ER owing to complications or non-curative resection were included. Moreover, patients who underwent ER alone for EGC without surgery between May 2005 and March 2016 were included and followed up for at least 5 years. After excluding patients with missing data, a total of 14,760 patients who underwent surgery (*n* = 12,631) or ER alone (*n* = 2129) were included (Figure 1). The patients were randomly divided into the development set (70%) and validation set (30%).

### 2.2. Definition, Outcome, Data Sources, and Study Variables

LNM was defined based on surgical specimens of patients who underwent surgery. In patients who underwent ER alone, regional LN recurrence was determined based on computed tomography scans during follow-up.

The outcome consisted of establishing the ML model for predicting LNM risk in EGC patients and validating its performance. We divided the entire cohort into a development set (70%) for derivation of the ML model and a validation set (30%) for validation. Since the actual target participants were patients treated with ER for EGC, the performance of the ML model was evaluated for total patients and initial ER patients, respectively, using three methods in the development set and validation set. First, the area under the receiver operating characteristic (AUROC), sensitivity, and specificity of the ML model were analyzed. Second, we assessed whether the ML model could stratify the risk of LNM into very low-, low-, intermediate-, and high-risk categories. In the development set, we listed the predicted values calculated by the ML model and selected cutoffs at the points where the actual LNM rates were 1%, 3%, and 7%. An actual LNM rate <1% was allocated into the very low-, <3% into the low-, <7% into the intermediate-, and ≥7% into the high-risk categories. The 3% and 7% criteria for the low-, intermediate-, and high-risk categories were based on the previous literature [12]. Additionally, we set a very-low risk category of predicted LNM risk with <1%. This ML model for stratifying LNM risk was applied to the total patients and patients with initial ER in the validation set. Third, we evaluated the ability of the ML model to discriminate patients with negligible risk of LNM at a high-sensitivity cutoff of 100% to predict LNM. From a clinical perspective, the utility of a risk score depends on its ability to discriminate patients at low risk for LNM, i.e., it is ideal to identify patients who do not need surgery and those who need surgery.

Non-curative resection was defined as not satisfying an expanded criterion for curative resection. The expanded criteria for curative resection were en bloc resection, negative horizontal and vertical margins, absence of lymphovascular invasion, and one of the following: (a) differentiated mucosal cancer without ulcerative lesions, regardless of the tumor size; (b) differentiated mucosal cancer with ulcerative lesions that were ≤3 cm in size; (c) undifferentiated mucosal cancer without ulcerative lesions that were ≤2 cm in size; or (d) differentiated cancer invasion to the submucosa <500 µm from the muscularis mucosa that was ≤3 cm in size.

Data were collected retrospectively from the electronic medical records, including age, sex, number of tumors, tumor location (upper third, middle third, and lower third), size (mm), gross type (non-depressed and depressed), differentiation (well, moderate, signet, and poor), Lauren classification (intestinal, diffuse, and mixed), depth of invasion (lamina propria, muscularis mucosa, submucosal invasion <500 µm from the muscularis mucosa (SM1), and submucosal invasion ≥500 µm from the muscularis mucosa (SM2/3)), lymphatic invasion, venous invasion, and perineural invasion.

### 2.3. Establishment of the Machine Learning Model

The ML model was implemented using 3 methods to produce an optimal model based on the development set (70%): logistic regression, support vector machine (SVM), and random forest (RF). We constructed the ML model in the cohort of total patients and patients with initial ER, respectively. This design considered our actual target as EGC patients who were feasible ER. A randomized search algorithm with fivefold nested cross-validation in the development set was conducted for hyperparameter optimization of each method. The algorithm was optimized by randomly searching the given hyperparameter space 1000 times using the development set (Appendix A). We selected this search algorithm rather than grid or Bayesian search algorithms because these three methods are fast enough to search all given spaces and have relatively few hyperparameters. The best hyperparameters in a model were chosen when the model had the highest AUROC. The performance of the models with the best hyperparameters was evaluated in the validation set (30%). We defined the weighted factors of 14.0 through the imbalanced rate of the classes. We confirmed the feature importance as permutating a specific variable 100 times. We publicly opened the codes and models at https://github.com/YeongChanLee/Predict-LNM (accessed on 21 February 2022).

### 2.4. Statistical Analysis

Baseline characteristics were compared between the development and validation sets and presented as means (standard deviation) and frequencies (%) for continuous and categorical variables, respectively. The performance of the ML model was evaluated using AUROC, sensitivity, and specificity. The sensitivity and specificity were derived using Youden’s index. The risk probability was calculated for the stratification of LNM risk based on the logistic regression, RF, and SVM analyses in the development set. Predicted LNM risk was classified into very low-, low-, intermediate-, and high-risk categories according to the actual LNM rate with a cutoff <1%, <3%, and <7%. We analyzed whether the categories of predicted LNM risk correlated with the real LNM rate. As a subanalysis, the performance of the ML model was compared with the eCura system as a clinical model in cases defined as non-curative resection after ER for EGC in the validation set, using AUROC, net reclassification improvement (NRI), and specificity at a high-sensitivity cutoff of 95%. The ML model was developed using Scikit-learn 0.24.1 and Python 3.8.5. Statistical analyses were performed using R (version 3.5.1, Vienna, Austria).

## 3. Results

### 3.1. Baseline Characteristics

A total of 14,760 patients were eligible for analysis; 10,332 patients were randomly sorted into the development set and 4428 into the validation set. LNM was found in 794 of 10,332 patients (7.7%) in the development set and 337 of 4428 patients (7.6%) in the validation set. The baseline characteristics of the development and validation sets are shown in Table 1. They were comparable in most variables, including age, sex, number of tumors, size, gross type, differentiation, Lauren classification, depth of invasion, lymphatic invasion, venous invasion, and perineural invasion. However, the middle-third of the stomach was the most frequent tumor location in the development set whereas the lower-third of the stomach was the most frequent tumor location in the validation set (*p* = 0.013).

### 3.2. Derivation of the Machine Learning Model

In the development set, LNM was found in 794 of 10,332 patients (7.7%) in the total patients, and in 42 of 2320 patients (1.8%) in patients with initial ER. The derivatated ML model showed good to excellent performance in the development set; in the total patients, logistic regression was AUROC (95% CI), 0.86 (0.85–0.88); sensitivity, 0.80; and specificity, 0.76; RF was AUROC (95% CI), 0.95 (0.94–0.95); sensitivity, 0.91; and specificity, 0.86; and SVM was AUROC (95% CI), 0.87 (0.85–0.88); sensitivity, 0.79; and specificity, 0.78. In patients with initial ER, logistic regression was AUROC (95% CI), 0.88 (0.83–0.92); sensitivity, 0.86; and specificity 0.82; RF was AUROC (95% CI), 0.95 (0.93–0.97); sensitivity, 0.93; and specificity, 0.88; and SVM was AUROC (95% CI), 0.88 (0.83–0.92); sensitivity, 0.93; and specificity, 0.73 (Figure 2).

In the development set, LNM risk was predicted using the ML model (logistic regression, RF, and SVM), and the cutoff for the categories of very low, low, intermediate, and high risk was set as the value of the actual LNM rate of <1%, <3%, and <7% in the total patients and initial ER patients, respectively (Table 2). As an example, in the total patients, LNM risk was stratified using logistic regression into very low (<1%)-, low (<3%)-, intermediate (<7%)-, and high (≥7%)-risk categories, and the cutoff was determined by the actual LNM rate. Each category showed a real LNM rate of 0.2%, 1.4%, 4.1%, and 18.4% (Table 2).

### 3.3. Validation of the Machine Learning Model

In the validation set, LNM was found in 337 of 4428 patients (7.6%) in the total patients, and in 24 of 1016 patients (2.4%) in patients with initial ER. In the validation set, the ML model showed a good performance in the total patients and patients with initial ER. In total patients, logistic regression was AUROC (95% CI), 0.86 (0.84–0.88); sensitivity, 0.80; and specificity, 0.75; RF was AUROC (95% CI), 0.85 (0.83–0.87); sensitivity, 0.82; and specificity, 0.72; and SVM was AUROC (95% CI), 0.86 (0.84–0.88); sensitivity, 0.69; and specificity, 0.85. In patients with initial ER, logistic regression was AUROC (95% CI), 0.90 (0.86–0.94); sensitivity, 0.92; and specificity, 0.77; RF was AUROC (95% CI), 0.88 (0.82–0.92); sensitivity, 0.92; and specificity, 0.74; and SVM was AUROC (95% CI), 0.89 (0.85–0.93); sensitivity, 0.92; and specificity, 0.78 (Figure 3).

In the validation set, logistic regression and SVM showed the possibility of stratifying the risk of LNM for total patients and patients with initial ER. The predicted LNM risk was correlated with the actual LNM rate. In the total patients, the actual LNM rate according to the very low-, low-, intermediate-, and high-risk categories was 0.1%, 1.6%, 4.8%, and 17.7% based on logistic regression and 0.1%, 1.6%, 4.2%, and 18.1% based on SVM, respectively. In patients with initial ER, the actual LNM rate according to the very low-, low-, intermediate-, and high-risk categories was 0.2%, 2.5%, 0.0%, and 11.9% based on logistic regression and 0.2%, 1.7%, 4.5%, and 13.0% based on SVM, respectively. In contrast, in the analysis using RF, the actual LNM rate was 1.3%, 6.3%, 7.4%, and 23.1% of the total patients and 0.4%, 5.0%, 10.0%, and 12.0% of patients with initial ER, which was higher than that of the predicted category of LNM risk (Table 3).

In the total patients in the validation set, the specificities of the ML model at the high-sensitivity cutoff of 100% were 49%, 46%, and 49% in the logistic regression, RF, and SVM analyses, respectively. In patients with initial ER, the specificities of the ML model at the high-sensitivity cutoff of 100% were 71%, 57%, and 70% in the logistic regression, RF, and SVM analyses, respectively (Figure 4).

In the validation set, as a subanalysis in the patients with non-curative resection after ER for EGC, LNM was found in 21 of 362 patients (5.8%). The AUROC of the ML model was 0.76, 0.73, and 0.75 in the logistic regression, RF, and SVM analyses, respectively, and the AUROC of the eCura system was 0.72. Logistic regression (NRI, 0.46) and SMV (NRI, 0.21) improved the performance compared to the eCura system. The specificities of the ML model at the high-sensitivity cutoff of 95% were 39%, 38%, and 38% in the logistic regression, RF, and SVM analyses, respectively, which were higher than the specificity of 9% for the eCura system (Appendix A).

## 4. Discussion

Here, we demonstrated the utility of an ML model for predicting the LNM risk in EGC patients. In the validation set, the AUROC of each ML model showed a good performance, ranging from 0.85 to 0.90. Furthermore, each ML model could stratify the LNM risk as very low, low, intermediate, and high risk, and those stratified groups showed a consistent actual LNM rate. In addition, these showed specificities of about 0.50 or higher at a matched sensitivity of 100%, indicating that it could discriminate patients with negligible risk of LNM while identifying the patients who needed surgery owing to the LNM risk with 100% sensitivity. This tool can easily be applied in clinical practice to categorize the LNM risk and identify patients with negligible LNM risk under the assumption of maximum sensitivity.

Non-curative resection after ER for EGC patients is a clinical concern. Physicians determine further strategies under careful consideration, accounting for the patient’s comorbidities associated with surgical risk and individual preference, and the characteristics of the tumor and surgical procedure. Despite additional surgery owing to non-curative resection after ER, the rate of LNM is only 5–10%; hence, among the patients with non-curative resection, it is clinically significant to identify patients at low risk of LNM to prevent unnecessary surgery. The current guidelines have been revised to address these issues and recommend a more detailed strategy after non-curative resection [1,2,4,11]. In the JGCA guidelines (5th edition), among the factors of non-curative resection, piecemeal resection or a positive lateral margin is defined as eCura C-1, and other factors are described as eCura C-2. Based on these classifications, physicians can determine the appropriate therapeutic options, such as additional ER or coagulation for patients in eCura C-1. For eCura C-2, the eCura scoring system was built based on large-scale data and stratifies LNM risk as low (0–1 point), intermediate (2–4 points), or high (5–7 points) [11,12]. In patients with the low-risk category, there is no difference in cancer recurrence or cancer-specific mortality between patients who receive no additional treatment and those who undergo additional surgery [14]. Similarly, reports that investigated LNM risk in patients with early colon cancer after ER were conducted to prevent unnecessary surgery or excess treatment using the AI system and clinical guidelines [24,25,26,27]. This reflects the necessity for detailed guidance on additional strategies through the stratification of LNM risk in EGC patients with non-curative resection after ER; therefore, this study has clinical significance.

The strength of this study is that it is the first to develop an ML model to predict LNM in patients with EGC and validate its good performance. Furthermore, our study was based on a large sample size and investigated three models (logistic regression, RF, and SVM) to develop an optimal ML model. Considering that the target participants were patients who underwent ER for EGC, the performance of the ML model was verified not only for the total patients but also the patients who received ER as the initial treatment for EGC. In our study, the very low-risk group had an LNM rate of <1%. This is a stricter category than the classifications of previous reports that defined a low risk of LNM as <3%, including nomograms and the eCura system for predicting LNM in EGC patients [11,28]. In addition to the variables included in the nomogram and the eCura system, our ML model was constructed based on various variables, including the number of tumors, tumor location, Lauren classification, perineural invasion, age, sex, gross type, tumor size, differentiation, depth of invasion, lymphatic invasion, and venous invasion [12,28]. Moreover, we utilized the ability of the ML model to comprehensively interpret various factors by subdividing the data of the variables assessed in previous reports [12,28]. For example, the depth of invasion was subdivided into the lamina propria, muscularis mucosae, SM1, and SM2/3.

We evaluated the performance of the ML model using clinically relevant outcomes. In estimating LNM risk in patients with non-curative resection after ER for EGC, achieving a high sensitivity to predict LNM is essential for long-term outcomes. Furthermore, there is a need to identify patients at low risk for LNM to prevent unnecessary surgery. Our ML model showed specificities of 49% in the total patients and 71% in the patients with initial ER at the high-sensitivity cutoff of 100%. When examining only patients with non-curative resection after ER, our ML model showed specificities ranging from 38% to 39% at the high-sensitivity cutoff of 95%, which is significantly increased compared to the specificity of 9% for the eCura system. The sensitivity of 95% was set based on the highest sensitivity achieved by the eCura system. Therefore, the ML model has great clinical potential in that it had better specificity than the eCura system at a high-sensitivity cutoff, despite there being no significant difference in the value of AUROC.

This study had several limitations. First, there may be selection bias due to the exclusion of missing data and the study’s retrospective nature; however, this study was designed to develop the ML model, including major factors without missing data. Second, this was a single-center study, and the results need to be validated in other institutions. In addition, it is necessary to validate the performance of the ML model in patients undergoing non-curative resection after ER for EGC. Through this additional validation, we can anticipate the improved version of the ML model by reinforcement learning and suggest that the ML model can be a valuable tool in clinical applications. Third, most of the variables included in our ML model are based on the pathology after ER. For estimation of LNM risk, several major variables, such as lymphatic invasion, vertical margin, and the depth of invasion, could not be assessed by endoscopy alone. Fourth, the comparison of long-term survival was not analyzed according to the stratification of LNM risk, as there were some cases with insufficient follow-up because the follow-up ended in March 2021.

In conclusion, the ML model showed good performance in the prediction and stratification of LNM risk in patients with EGC. Based on this finding, we suggest that the ML model has the potential to be a clinically useful tool for estimating LNM risk among patients with non-curative resection after ER.

## Figures and Tables

**Figure 1 cancers-14-01121-f001:**
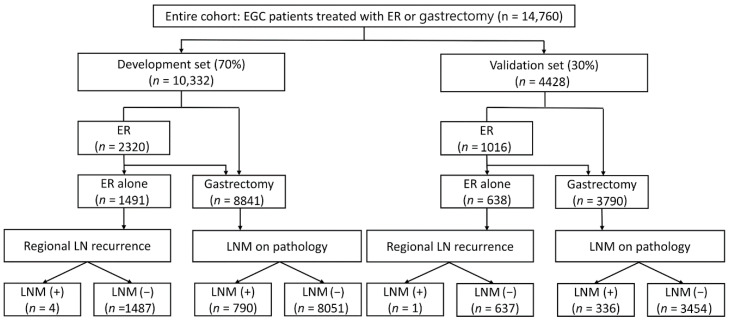
Diagram of patient selection.

**Figure 2 cancers-14-01121-f002:**
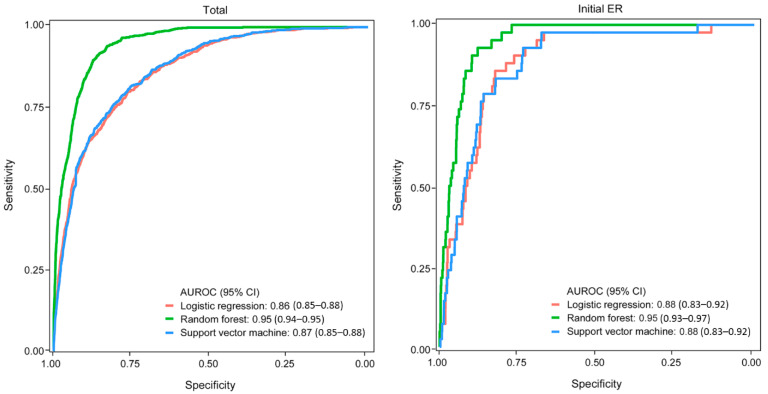
AUROC of the ML model for the prediction of LNM in the development set (total number = 10,332, number of patients with initial ER = 2320).

**Figure 3 cancers-14-01121-f003:**
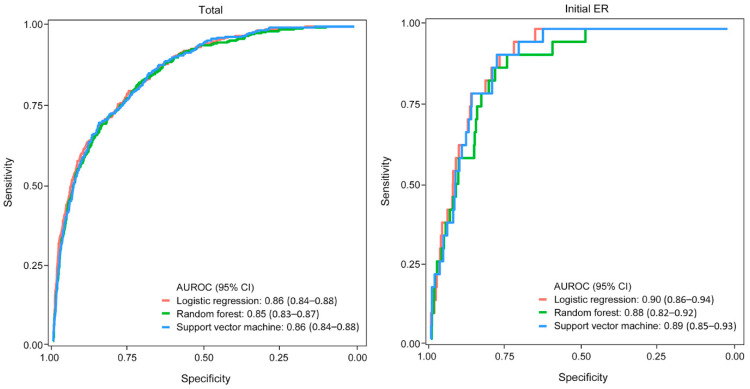
AUROC of the ML model for the prediction of LNM in the validation set (total number = 4428, number with initial ER = 1016).

**Figure 4 cancers-14-01121-f004:**
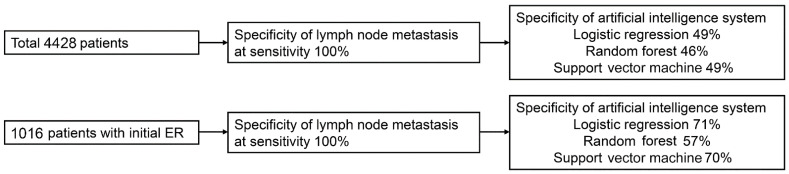
Identification of patients with negligible risk of lymph node metastasis at the high-sensitivity cutoff in the validation set.

**Table 1 cancers-14-01121-t001:** Baseline characteristics of the development set and validation set.

Variable	Development (*n =* 10,332)	Validation (*n =* 4428)	*p* Value ^a^
Age ^**†**^	58 ± 11	58 ± 11	0.413
Gender			0.789
Male	6697 (65)	2881 (65)	
Female	3635 (35)	1547 (35)	
tumors	512 (5)	201 (5)	
Location			0.013
Upper	1083 (11)	483 (11)	
Middle	4773 (46)	1929 (44)	
Lower	4476 (43)	2016 (45)	
Size (mm) ^**†**^	27 ± 18	27 ± 18	0.645
Gross type			0.823
Non-depressed	2568 (25)	1109 (25)	
Depressed	7764 (75)	3319 (75)	
Differentiation			0.999
Well	1214 (12)	523 (12)	
Moderate	4053 (39)	1741 (39)	
Signet	2315 (22)	989 (22)	
Poorly	2750 (27)	1175 (27)	
Histologic type by Lauren			0.122
Intestinal	5198 (50)	2271 (51)	
Diffuse	3867 (38)	1666 (38)	
Mixed	1267 (12)	491 (11)	
Depth of invasion			0.983
Lamina propria	2568 (25)	1114 (25)	
Muscularis mucosa	3767 (37)	1612 (37)	
SM1	1069 (10)	455 (10)	
SM2/3	2928 (28)	1247 (28)	
Lymphatic invasion, present	1571 (15)	682 (15)	0.780
Venous invasion, present	154 (2)	72 (2)	0.588
Perineural invasion, present	232 (2)	96 (2)	0.817

^**†**^ Mean ± standard deviation presented for continuous variables. Values are expressed as *n* (%); unless otherwise specified. ^**a**^
*p*-value calculated using Student’s *t*-test for continuous variables or Pearson’s chi-square test for categorical variables for overall data. SM1: submucosal invasion <500 µm from the muscularis mucosa; SM2/3: submucosal invasion ≥500 µm from the muscularis mucosa.

**Table 2 cancers-14-01121-t002:** Determination of the cutoff for stratification of LNM risk based on the predictive value of the ML model and actual LNM rate in the development set. (**A**) Total patients. (**B**) Patients with initial ER.

**(A) Total Patients (*n* = 10,332) and LNM (*n* = 794)**				
Logistic regression				
*n* of patients	*n* of LNM	Rate (%)	Risk probability	Risk category
1863	3	0.2	<1%	Very low
3105	42	1.4	≥1% to <3%	Low
1656	67	4.1	≥3% to <7%	Intermediate
3708	682	18.4	≥7%	High
Random forest				
*n* of patients	*n* of LNM	Rate (%)	Risk probability	Risk category
5589	2	<0.1	<1%	Very low
1859	24	1.3	≥1% to <3%	Low
412	18	4.4	≥3% to <7%	Intermediate
2472	750	30.3	≥7%	High
Support vector machine				
*n* of patients	*n* of LNM	Rate (%)	Risk probability	Risk category
2277	5	0.2	<1%	Very low
2691	35	1.3	≥1% to <3%	Low
1656	65	3.9	≥3% to <7%	Intermediate
3708	689	18.6	≥7%	High
**(B) Initial ER** **(*n* = 2320) and LNM (*n* = 42)**				
Logistic regression				
*n* of patients	*n* of LNM	Rate (%)	Risk probability	Risk category
1492	1	0.1	<1%	Very low
368	5	1.4	≥1% to <3%	Low
92	3	3.3	≥3% to <7%	Intermediate
368	33	9.0	≥7%	High
Random forest				
*n* of patients	*n* of LNM	Rate (%)	Risk probability	Risk category
1722	0	0	<1%	Very low
322	4	1.2	≥1% to <3%	Low
46	2	4.4	≥3% to <7%	Intermediate
230	36	15.7	≥7%	High
Support vector machine				
*n* of patients	*n* of LNM	Rate (%)	Risk probability	Risk category
1491	1	0.1	<1%	Very low
136	2	1.5	≥1% to <3%	Low
445	15	3.3	≥3% to <7%	Intermediate
206	24	10.4	≥7%	High

LNM, lymph node metastasis.

**Table 3 cancers-14-01121-t003:** Risk stratification of LNM by the ML model and the actual rate in the validation set. (**A**) Total patients. (**B**) Patients with initial ER.

**(A) Total Patients (*n* = 4428) and LNM (*n* = 337)**				
Logistic regression				
Risk probability	Risk category	*n* of patients	*n* of LNM	Rate (%)
<1%	Very low	801	1	0.1
≥1% to <3%	Low	1335	21	1.6
≥3% to <7%	Intermediate	708	34	4.8
≥7%	High	1584	281	17.7
Random forest				
Risk probability	Risk category	*n* of patients	*n* of LNM	Rate (%)
<1%	Very low	2403	30	1.3
≥1% to <3%	Low	793	50	6.3
≥3% to <7%	Intermediate	176	13	7.4
≥7%	High	1056	244	23.1
Support vector machine				
Risk probability	Risk category	*n* of patients	*n* of LNM	Rate (%)
<1%	Very low	978	1	0.1
≥1% to <3%	Low	1138	19	1.6
≥3% to <7%	Intermediate	678	30	4.2
≥7%	High	1297	287	18.1
**(B) Patients with Initial ER (*n* = 1016) and LNM (*n* = 24)**				
Logistic regression				
Risk probability	Risk category	*n* of patients	*n* of LNM	Rate (%)
<1%	Very low	656	1	0.2
≥1% to <3%	Low	160	4	2.5
≥3% to <7%	Intermediate	40	0	0
≥7%	High	160	19	11.9
Random forest				
Risk probability	Risk category	*n* of patients	*n* of LNM	Rate (%)
<1%	Very low	756	3	0.4
≥1% to <3%	Low	140	7	5.0
≥3% to <7%	Intermediate	20	2	10.0
≥7%	High	100	12	12.0
Support vector machine				
Risk probability	Risk category	*n* of patients	*n* of LNM	Rate (%)
<1%	Very low	655	1	0.2
≥1% to <3%	Low	59	1	1.7
≥3% to <7%	Intermediate	191	9	4.5
≥7%	High	87	13	13.0

## Data Availability

The data presented in this study are available on request from the corresponding author. The data are not publicly available due to personal privacy.

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
