# Peer review of "Machine Learning Model to Stratify the Risk of Lymph Node Metastasis for Early Gastric Cancer: A Single-Center Cohort Study"

_cancers, 2022, doi:10.3390/cancers14051121_

Round 1
Reviewer 1 Report
The authors have satisfactorily addressed all my concerns and made the necessary changes to the manuscript. The work is really impressed with a very large-scale cohort and well-designed experiments. I would suggest the manuscript to be accepted. Incidentally, besides the clinical characteristics used in this study, some novel technologies like AI/Radiomics are also extensively studied for LNM prediction in gastric cancer. Such as the first radiomics studies about LNM in gastric cancer: 10.1016/j.annonc.2020.04.003; 10.3389/fonc.2019.00340; 10.1007/s00330-019-06398-z; and 10.1007/s00330-019-06621-x. A short discussion or outlook of these new technologies may bring new research ideas to the oncologists in gastrointestinal tumor.Author Response
We thank the reviewer for this encouraging comment.
Reviewer 2 Report
I agree with the revisions made by the authors in the revised version.Author Response
Thank you sincerely for your review.
Reviewer 3 Report
I would like to thank the authors for their detailed responses. However, my problem is still there, the group underwent gastrectomy directly (referred as gastrectomy alone group) is different from ER group (ER alone or ER+gastrectomy) by nature. The group underwent gastrectomy directly would be in advanced stage, while ER groups in earlier stage. Thus, if the authors intend to claim the use of ML model in EGC under ER, the gastrectomy alone group should be excluded both in development set and validation set. In the manuscript, I found that the majority cases were gastrectomy alone group (8012 in development set and 3412 in validation set). Based on the ML algorithms, the ML models were predominated by the majority group. Without doing this, the authors were using model trained by gastrectomy alone group (advanced stage) to apply on ER groups (early stage). That is not reasonable.
To use pathologic features after ER as the input, the issue also lies on gastrectomy alone group. Due to ER not done for the gastrectomy alone group, how possible the authors able to input pathologic features that were obtained from ER?
Round 2
Reviewer 3 Report
The authors have addressed my concerns quite well. I don’t have further questions and feel like to endorse the publication.
This manuscript is a resubmission of an earlier submission. The following is a list of the peer review reports and author responses from that submission.
Round 1
Reviewer 1 Report
This paper presents a machine learning model to stratify the risk of lymph node metastasis (LNM) for early gastric cancer (EGC). The authors used clinical variables and basic machine learning algorithms to classify the LNM risks. However, their approach is limited on novelty and improvement compared with related studies. I have some concerns.
1. In part 4 Discussion, the authors stated, “This study is the first to create an ML model to predict LNM in patients with EGC and validate its good performance.” However, as far as I know, predicting LNM risks has been explored in at least the following literature.
Dual-energy CT–based deep learning radiomics can improve lymph node metastasis risk prediction for gastric cancer
Deep learning analysis of the primary tumour and the prediction of lymph node metastases in gastric cancer
The former treated all lymph nodes as a whole object, while the latter predicted the risks for each separate lymph node station. Both of the two approaches utilized deep learning models on medical images. In this paper, the author used clinical characters such as “Depth of invasion”, “SM1”, “SM 2/3”, etc., which are also based on visual observations on medical images. Meanwhile, the performance of these two papers reached a relatively high sensitivity and specificity at the same time.
The authors need to elaborate on the difference, novel contribution, and performance improvement compared with existing studies.
2. In part 2.2, the authors came up with two outcomes: 1) stratification of LNM risks; 2) discrimination of patients with negligible risk of LNM. In the result parts:
Task 1 results in table 2 seem like a regression problem to stratify the patients into ten deciles and categorize them into four risk groups. Patients of 10 deciles have nearly the same number. I wonder if it is the result of the model prediction. Also, the risk splitting criteria need to be clearly stated. Is it based on the number of LNM? The meaning of the table is unclear to me.
For task 2 results in figure 2, the AUC curves are strange with paintbrush lines, which seems informal in a research paper. It looks like a classification problem. So, what is the relationship with task 1? Is it separate or combined? Also, in the discussion part, the author stated, “In addition, it showed a specificity of 26% at a matched sensitivity of 100%, which means that it could predict patients who needed surgery owing to the LNM risk with 100% sensitivity.” Any binary classifier can reach a sensitivity of 100% at a certain threshold. It is meaningless to discuss the sensitivity with no attention to specificity. I don’t think a specificity of 26% is a good performance for medical purposes since there might be a lot of overdiagnosis.
3. In figure 1, in ER alone data, the LNM positive patients have a super small number of 5, compared with the entire cohort of 14 760. It seems unreasonable to mix the ER alone patients with gastrectomy patients considering the significant distribution difference. It might be better if the authors could mention the model performance for ER alone patients.
4. The authors used relatively simple and traditional algorithms, and the analysis of the predicted results is also limited. It might be better to include more experiments and discussions on the interpretability of model decisions, such as analyzing the variable importance for predicting LNM risk.
5. The data source needs to be clearly stated. Is it from a single hospital or multiple centers? If it is a single-center study, there might be a lack of external validation.
Reviewer 2 Report
The study by Na et al about early gastric cancer and how to stratify the risk of lymph node metastasis by using machine learning is well written and clearly described.
The paper describes the challenges with endocopic resection and overtreatment and suggest machine larning as one way to overcome this problem.
The topic is clinical cancer diagnostics and treatment strategies. The topic includes machine learning which is very up to date and a field that is under extensive development today.
The paper adds the use of machine learning to the current field. Even if this is a grwoing field, it is not fully used or developed. Which this paper shows. The authors do claim several limitations, but are very claer about those, and have suggestions of how to validate those further in future studies. This is an important statement and shows that this is a novel field in cancer diagnostics.
Otherwise, the authors state themselves that limitations such as modelling the mahine learning approach in patients undergoing non-curative resection after endoscopic resection and some variables that are not included in the model. Moreover, the authors state that long-term survival was not analyzed according LNM risk stratification due to insufficient follow-up because of the date ended March 2021.
The conlusions may be too early to draw since many limitations are stated. However, since the authors are very clear about the limitations, I think it is fine as a contribution to this field under development.Therefore, the main questions are posed.
I have some minor comments:
The authors claim that there are limitations with the study. The title could be changed to include also the term ”single-center study”.
Reviewer 3 Report
The authors proposed a LNM predictive model for ECG. The attempt is attractive, however, several flaws and technical issues need substantial improvement. My major concerns are list as following. On M&M Many sources of bias should be carefully addressed, I list them as following 1) According to the intended use of the LNM predictive model proposed by the authors, only ECG patients who received ER are eligible. Enrollment of ECG patients who got gastrectomy is in conflict with the goal of predicting LNM in ECG patients underwent ER. Based on the composition of the cases, it is obviously the predictive model was majorly affected by the data of ECG cases underwent gastrectomy. In brief, the model was to predict LNM in gastrectomy cases but absolutely not in ER cases. 2) Many input features were pathologic data, which means the input features were generated together with the status of LNM. Thus, when the input features and the answer of LNM can be found in the pathology report, why should we input the features into a model and get the LNM (we have already known that in pathology report!)? 3) One-time training and validation produces strongly biased performance estimates. Typically it takes three steps: training, validation, and independent testing processes. Or Nested cross validation and train/test split approaches produce robust and unbiased performance estimates. For using nested cross validation, please refer to the publication (https://doi.org/10.1371/journal.pone.0224365). Moreover, I suggest reporting the 95% CI for all the performance metrics as well as the ROC curves.